

# The impact of climate change on dam overtopping flood risk

Michelle Ho[1], Declan O'Shea[1,2], Conrad Wasko[3], Rory Nathan[1], Ashish Sharma[4]

[1]Department of Infrastructure Engineering, The University of Melbourne, Melbourne, 3010, Australia

[2]HARC-Hydrology and Risk Consulting, Melbourne, 3130, Australia

[3]School of Civil Engineering, The University of Sydney, Sydney, 2050, Australia

[4]School of Civil and Environmental Engineering, The University of New South Wales, Sydney, 2052, Australia

*Correspondence to*: Michelle Ho (m.ho@unimelb.edu.au)

**Abstract.**

There is unequivocal evidence that climate change will change the risk profile of dams, which are critical pieces of
infrastructure that safeguard water supply and provide flood mitigation for populated areas. However, the challenges involved in estimating the probability of extreme floods under climate change have meant that few studies have estimated the plausible changes in the risk of extreme floods that have the potential to overtop dams. A recent examination of contemporary scientific findings pertinent to climate change impacts on flood risk has informed the projection of extreme flood risk and dam overtopping risk estimates made here. We project changes in the exceedance probabilities of overtopping risk for 18 large
dams in Australia under a range of global warming assumptions, where consideration is given to the impacts of climate change on rainfall depth, rainfall temporal pattern, and rainfall losses resulting from changes in antecedent catchment wetness. We used event-based flood modelling and Monte Carlo sampling to appropriately represent the range of uncertainties associated with projecting estimates of extreme flood risk. Our results are presented in terms of changes per degree of global warming, which facilitates their interpretation in terms of different greenhouse gas emission scenarios and future time horizons. We
found that increases in rainfall depth had the largest impact on increasing dam overtopping flood risk for all 18 dams under climate change. Under 4ºC of global warming, which approximates conditions towards the end of this century under a high emissions scenario, the risk of overtopping floods was between 2.4-17 times that of historical conditions for the 18 dams investigated. We also found that the risk of overtopping has more than doubled compared to the historical baseline for four of the dams investigated here as a result of global warming that has already occurred.




**Graphical abstract**

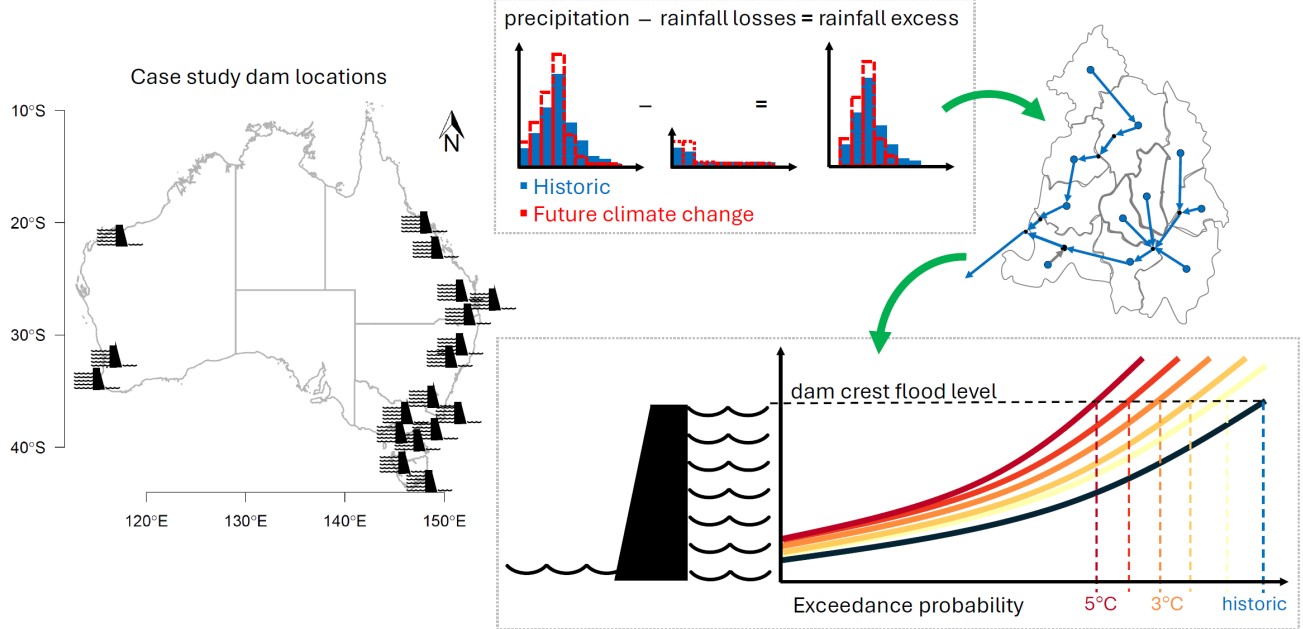

## 1 Introduction

The number of flood disasters has risen, more than doubling in the last two decades (2000-2019 compared with 1980-1999)
(Yaghmaei et al., 2020), and associated financial losses over the past five years have amounted to $320 billion (USD) (Munich
Re, 2024), which are expected to continue increasing with global warming (Wasko et al., 2021b). The estimation of extreme
flood risk is therefore essential for managing flood responses and mitigation strategies including planning, design, and
management of infrastructure, emergency responses, and the setting of insurance premiums. The changing risk of rare floods
under climate change is of particular concern with respect to large high-risk infrastructure such as nuclear power plants (Prasad
et al., 2011) and large dams (Nathan and Weinmann, 2019a), where failures would threaten lives, livelihoods, and facilities
integral to supporting economic activity. The theoretical basis for estimating flood risk under a stationary climate is a relatively
mature science and a degree of consensus is reflected in national guidelines for flood risk estimation that are widely used in
practice throughout many parts of the world (Wasko et al., 2021a). However, it has long been recognised that global warming
is changing the hydrological cycle (e.g. Mitchell, 1989; Trenberth, 1999) and hence changing flood risk (Barnett et al., 2008;
Matalas, 1997).





There is irrefutable evidence that climate change has already impacted on elements that drive floods such as the frequency, intensity, and duration of rainfalls (Emori and Brown, 2005; Kunkel et al., 2013; Trenberth et al., 2003), with further changes projected to occur in the future. In addition to shifting the depth, location, and timing of moisture delivered during a flood event, changes in rainfall patterns also alter catchment moisture stores (Ho et al., 2022; Wasko et al., 2020; Woldemeskel and Sharma, 2016), which impact the subsequent flood response (e.g. Garg and Mishra, 2019; Ivancic and Shaw, 2015; Massari et

al., 2023; Sivapalan et al., 2005). The impact of climate change on floods has been widely recognised in the scientific literature (Bates et al., 2008; Kundzewicz et al., 2014). However, quantifying future flood risk is an ongoing challenge due to the compounding effects of aleatory (e.g. natural variability), epistemic (e.g. knowledge-based), and deep (e.g. climate change) uncertainties. Translating the available knowledge of climate change impacts on floods into guidance to inform practical applications for estimating flood risk, particularly extreme flood risk, is therefore relatively immature (Wasko et al., 2021a).

Much of the scientific literature pertaining to the impact of climate change on floods is focused on non-stationary flood frequency analysis (Salas et al., 2018; Stedinger and Griffis, 2011). However, non-stationary flood frequency approaches accounting for climate change have not been widely adopted in industry guidelines due to limited findings of robust and meaningful covariates for informing non-stationarity (Faulkner et al., 2020; Wasko et al., 2021a). Another approach widely used in the scientific literature is the "chain-of-models" approach, where climate projections from general circulation models

are downscaled and bias-correct to create local inputs for flood analysis (Hakala et al., 2019). While results from studies using a chain-of-models approach have been adopted in some flood estimation guidelines (e.g. Flood risk assessments: climate change allowances, 2024; Natural Resources Wales, Welsh Government, 2022; Willems, 2013), the method involves the propagation of cascading uncertainties. Consequently, existing guidelines for assessing the impacts of climate change on extreme floods either overly simplify the complexities of estimating the risks involved, or are dependent on methods that that

are too uncertain to justify their adoption in practice (Wasko et al., 2021a).

Many studies have acknowledged climate change as a source of increased risk to dams and the research focus has largely been on informing operational rules or adaptive management in the context of long-term changes in water supplies and demands (e.g. Fluixá-Sanmartín et al., 2021; Madani and Lund, 2010; Malerba et al., 2022; Tanaka et al., 2006). Some of these studies have included the consideration of a wide scope of climate change induced risks (e.g. changes in sedimentation rates, changes

in water demands, and changes in population exposure) (Fluixá-Sanmartín et al., 2019), without explicitly quantifying changes in dam overtopping risk. These studies used a chain-of-models approach resulting in projections of risk that range several orders of magnitude due to differences between general circulation model outputs. In contrast, examinations of climate change impacts on dam overtopping risk based on historical records have been based on the detection of trends in overtopping risk (Ahmadisharaf and Kalyanapu, 2015) or the prevailing hydroclimatology (Hwang and Lall, 2024). To date, there are a minimal

number of studies quantifying the impact of climate change on dam overtopping risk. One such study by Lee and You (2013)





provided a conceptual example intended for exploring the relative sensitivities of dam overtopping risk to changes in rainfall and reservoir capacity with time under climate change. As a result, uncertainties in the runoff response were not considered and the rates of change used to represent climate change were neither explicitly linked with scenarios of climate change nor global warming. Another study by Lompi et al. (2023) considered climate change using downscaled outputs from 12 climate

models under two emission scenarios in a chain-of-models approach. There is an imperative for dam owners to better understand the change in risk of extreme floods with the potential to overtop dams given the risk to downstream communities and industries dependent on the reservoir storage, as well as the potential for dams to be a device for mitigating climate change impacts (Boulange et al., 2021).

Our analysis is focussed on assessing the shift in the likelihood that dams will be overtopped by floods in a warming climate.

We investigate the performance of 18 large water-supply dams across Australia, which span different climates and catchment sizes. The analysis is based on the use of event-based flood modelling implemented within a stochastic framework as this is an approach that is well suited to explicitly considering the impacts of global warming on the salient flood drivers. Specifically, this includes the consideration of changes in rainfall intensities with temperature over a range of event magnitudes up to and including estimates of the Probable Maximum Precipitation (PMP) (Jakob et al., 2009; Visser et al., 2022); the rates of change

in storm temporal patterns with temperature (Visser et al., 2023); and changes in catchment antecedent wetness (Ho et al., 2022, 2023). The impacts of climate change on these flood drivers are considered both individually and in combination, and for a range of different degrees of global warming. We use a baseline time period of 1961-1990, which is herein referred to as the historic period. The historic period approximates the mid-point for much of the information used to derive the design information provided in Australian Rainfall and Runoff (the national flood guidelines for Australia), which establishes a

baseline of historic flood risk with which to compare climate change impacts. This differs from the 1850-1900 pre-industrial baseline period relevant to the Paris Agreement resulting in a difference of approximately 0.3°C of global warming between the pre-industrial baseline period and the 1961-1990 historic period used here.

## 2  Materials and methods

### 2.1  Case study locations

The 18 dams assessed in this study are owned and managed by nine major water agencies and utilities who are responsible for the largest dams in Australia. These 18 dams are primarily water supply dams and are all classified as large dams with wall heights ranging from 16-166 m. The catchments upstream of these dams range from 28-15,300 km$^2$ and are located across arid, temperate, and tropical climate zones (see Fig. 1 and Table 1). The case study dams are distributed across the Australian continent with the majority located in the more populous temperate climate zones. Together, these dams are subject to a diverse

range of extreme storm mechanisms as distinguished by their classification between different zones used for estimating the





probable maximum precipitation (PMP zones). Australia is divided into five PMP zones with the most prominent division being that of areas impacted by tropical storms, which are included in the Revised Generalised Tropical Storm Method (GTSMR – coastal and south-west Western Australia (SWWA)), and the south east of the continent, which is covered by the Generalised Southeast Australia Method (GSAM – coastal and inland) (see Fig. 1).

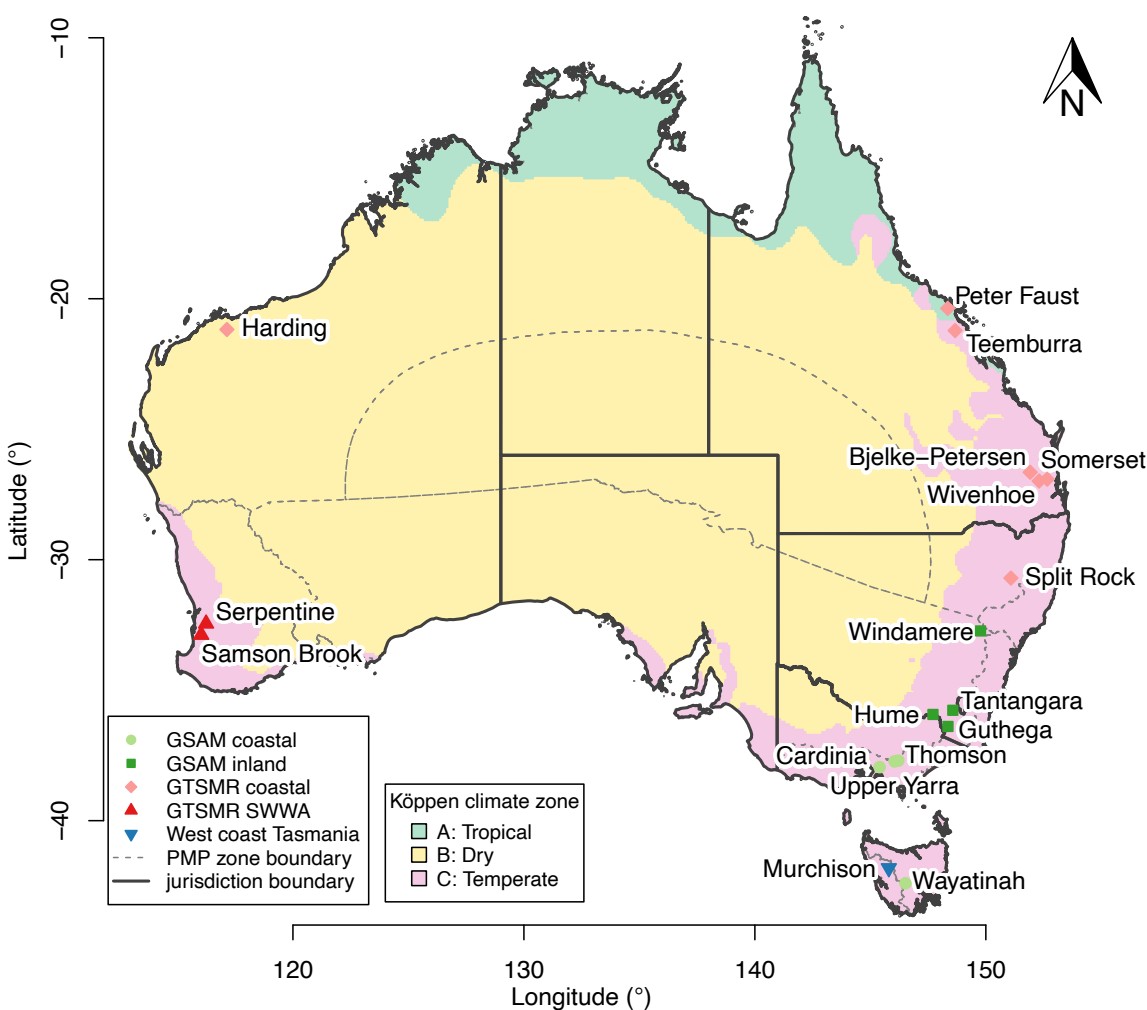


**Figure 1: Location of the 18 dams used for estimating flood risk under climate change in Australia and the associated zones used for estimating probable maximum precipitation.**





**Table 1: Catchment sizes, climate zones, PMP zones, and Natural Resource Management (NRM) regions of dam sites used for estimating flood risk under climate change.**

| Dam owner | Dam name | Area (km²) | Köppen Climate zone | PMP zone | State Jurisdiction | NRM region |
|---|---|---|---|---|---|---|
| Murray Darling Basin Authority | Hume | 15300 | Cfa | GSAM inland | NSW | Murray Basin |
| Snowy Hydro | Guthega | 91 | Cfb | GSAM inland | NSW | Southern Slopes |
| | Tantangara | 460 | Cfb | GSAM inland | NSW | Murray Basin |
| WaterNSW | Windamere | 1109 | Cfa | GSAM inland | NSW | Central Slopes |
| | Split Rock | 1618 | Cfa | GTSMR coastal | NSW | Central Slopes |
| Dept. of Regional Development, Manufacturing and Water | Peter Faust | 270 | Aw | GTSMR coastal | QLD | Wet Tropics |
| | Teemburra | 66 | Cwa | GTSMR coastal | QLD | Wet Tropics |
| Seqwater | Somerset | 1340 | Cfa | GTSMR coastal | QLD | East Coast |
| | Wivenhoe | 7020 | Cfa | GTSMR coastal | QLD | East Coast |
| Sunwater | Bjelke-Petersen | 1670 | Cfa | GTSMR coastal | QLD | East Coast |
| HydroTas | Murchison | 735 | Cfb | West coast Tasmania | TAS | Southern Slopes |
| | Wayatinah | 2130 | Cfb | GSAM coastal | TAS | Southern Slopes |
| Melbourne Water | Cardinia | 28 | Cfb | GSAM coastal | VIC | Southern Slopes |
| | Thomson | 487 | Cfb | GSAM coastal | VIC | Southern Slopes |
| | Upper Yarra | 337 | Cfb | GSAM coastal | VIC | Southern Slopes |
| Water Corporation (Western Australia) | Harding | 1071 | BWh | GTSMR coastal | WA | Rangelands |
| | Samson Brook | 64 | Csb | GTSMR SWWA | WA | Southern and South-Western Flatlands |
| | Serpentine | 665 | Csb | GTSMR SWWA | WA | Southern and South-Western Flatlands |

Köppen Climate zone abbreviations: Aw: equatorial, dry winter; BWh: arid, desert, hot; Cfa: warm temperate, fully humid hot summer; Cfb: warm temperate, fully humid, warm summer; Csb: warm temperate, dry, warm summer; Cwa: warm temperate, dry winter, hot summer. Jurisdiction abbreviations: NSW: New South Wales; QLD: Queensland; TAS: Tasmania; WA: Western Australia.

### 2.2  Event-based modelling

Flood exceedance probabilities were derived using event-based modelling within a Monte-Carlo framework. Event-based models were used as this method is best suited for both estimating extreme floods as well as explicitly accounting for climate change (Wasko et al., 2024a) while Monte-Carlo sampling allows for probabilistic sampling of the joint probabilities of flood inputs (Filipova et al., 2019; Kuczera et al., 2006; Nathan and Weinmann, 2019b). A schematic of the event-based modelling





process is shown in Fig. 2. The event-based runoff and streamflow routing procedures used in RORB (Laurenson, 1964;
Laurenson et al., 2010; Mein et al., 1974) were adopted here and emulated in the R software environment, referred to as
R$^2$ORB. This emulator handles data inputs, performs calculations, and generates outputs in a bespoke manner that enabled the
analysis to focus on the aspects of flooding most relevant to exploring climate change impacts. For each dam, the contributing
catchment was modelled as a semi-distributed conceptual node-link model. The catchments were divided into sub areas,
ranging in number from 4-19 subareas across the 18 case studies, to represent the stream network that allowed for rainfall to
be spatially distributed.

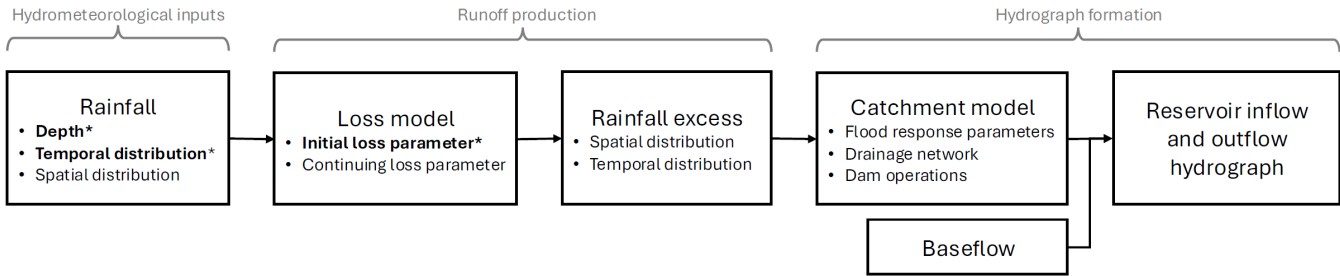

*inputs shown in **bold** are adjusted for climate change and stochastically sampled in a Monte Carlo framework

**Figure 2: Schematic of event-based flood modelling showing data inputs, models, model parameters, calculated outputs, and estimated hydrographs.**

For each event, we assumed that the reservoir was at a full supply level prior to the storm and only assessed flood events
resulting from the critical duration storm, that is, the storm duration identified in previous dam assessments conducted by the
dam owners as the storm duration that produced the largest reservoir outflows for extreme storms. We adopted the rainfall
spatial distribution patterns used by the dam owners and these were fixed for each Monte Carlo simulation. The rainfall excess
was calculated for each sub area using the initial loss continuing loss (ILCL) model and data inputs of rainfall depth, temporal
distribution, and spatial distribution. The rainfall excess was then routed through the catchment model to calculate the outflow
hydrograph. The outflow hydrograph was simulated 20,000 times by stochastically sampling rainfall depths, the initial loss
parameter, and the rainfall temporal distribution in order to derive the outflow flood frequency curve. (See Appendix A for
more details on the rainfall runoff model, the ILCL model, sampling approach, and derivation of the outflow flood frequency
curve.) The R$^2$ORB models were configured to reproduce inflow and outflow hydrographs and flood frequency curves
produced by the dam owners.

## 2.3 Assessing impacts of climate change

Climate change impacts on floods were examined by comparing outflow flood frequency curves derived from event-based
modelling using information on rainfall depth, storm temporal patterns, and rainfall losses, as described above and in Appendix





A. Investigations into climate change impacts on the spatial distribution of rainfall have shown that the spatial extents of storms are changing (e.g. Chang et al., 2016; Ghanghas et al., 2023, 2024; Lochbihler et al., 2017; Wasko and Sharma, 2017) but

results are as yet inconclusive and storm spatial patterns are therefore unchanged in this study. We report the impacts of climate change in terms of the relative shift in the annual exceedance probability (AEP) of the Dam Crest Flood (DCF) computed under historic (1961-1990) and future climatic conditions. The DCF is the flood event which, when routed through the dam storage, results in a peak water level that just reaches the crest level of the dam. The exceedance probability of the DCF is thus indicative of the probability that the dam is overtopped by a flood. Reporting climate impacts in terms of the relative shift in

the overtopping risk provides a non-dimensional metric that facilitates comparison across dams of different sizes and configurations, though it should be stressed that this metric should not be directly equated with the risk of dam failure as dams vary in their ability to accommodate overtopping for different depths and durations.

We calculated the relative shift ($RS$) in the risk of overtopping as follows:

$$RS = \frac{AEP_{DCF,p}}{AEP_{DCF}} \qquad (1)$$

where $AEP_{DCF,p}$ is the projected annual exceedance probability of the notional DCF under increased mean global temperature,

and $AEP_{DCF}$ is the annual exceedance probability of the DCF under historic conditions. The metric indicates the projected change in overtopping risk due to climate change, where a value of RS larger than 1 indicates an increased overtopping risk, while values less than 1 represent a decreased overtopping risk. For example, if the AEP of the notional DCF under historic climatic conditions is estimated to be 1 in 1,000,000, and the corresponding AEP under future climate is estimated to be 1 in 500,000, the relative shift in risk (RS) is 2.0; that is, the risk of a flood overtopping a dam is projected to double for the adopted

climate scenario. Conversely, if the estimated AEP of the DCF under climate change is 1 in 2,000,000, then overtopping risk halves (i.e. RS = 0.5).

The impacts of climate change were assessed using rates of change (or uplift factors) that varied with the degree of global warming, as applied to storm depth, temporal patterns and initial losses. Our assessment of climate change impacts was conditioned upon changes in mean global temperature as this is the primary driver of changes in atmospheric circulation and

moisture availability, which is also well simulated in general circulation models (Graham, 1995). Assessing the impacts of climate change with respect to increases in global temperatures also enables results to be translated to scenarios of climate change, future time horizons, and associated rates of global warming that are of interest to the dam owners. For example, our results based on a 4°C increase in mean global temperature approximates a high emissions scenario towards the end of the 21st century (see Fig. 3). The rates of changed used here to represent climate change impacts are consistent with the information

provided from a systematic review, metanalysis, and summary by Wasko et al. (2024a, b) and were first assessed individually,




and then in combination in response to increases in global temperature of 1-5°C in 1°C increments. The range of global warming explored was chosen to facilitate the interpretation of the results under a variety of global climate change scenarios and commonly considered future time horizons. The rationale for the rates of change used to adjust the flood drivers under global warming are provided below.

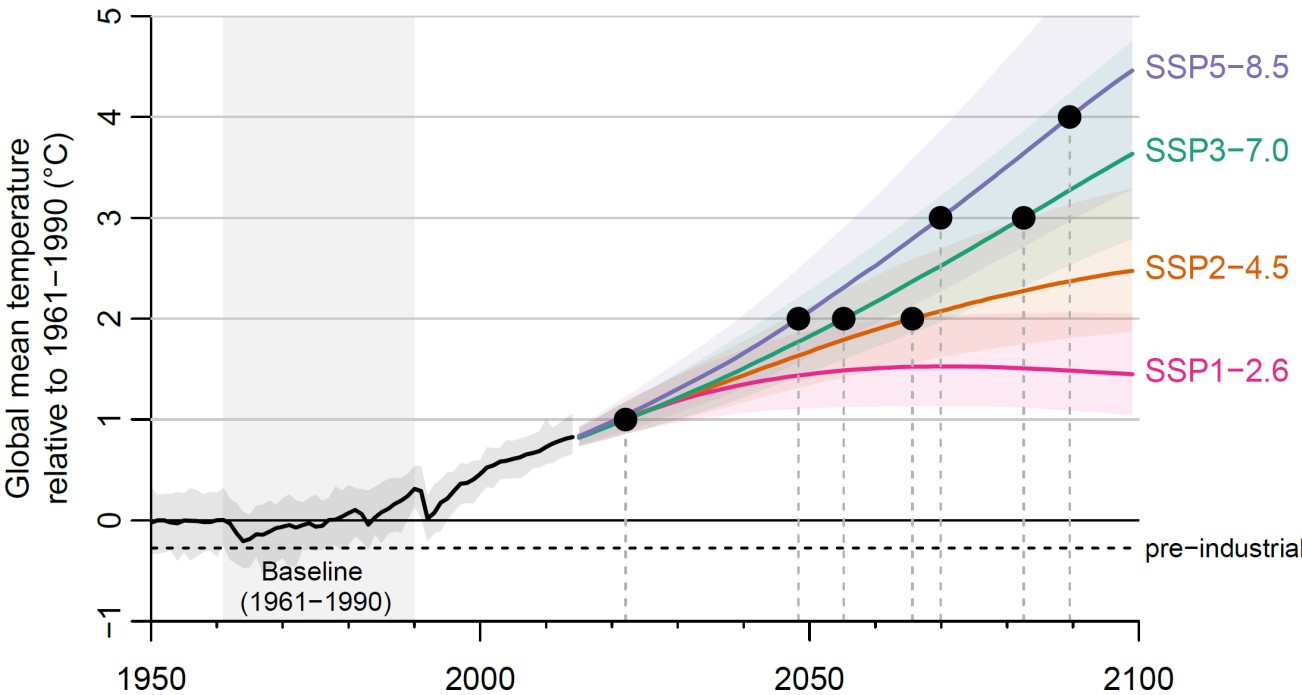


**Figure 3: Projected temperature increases associated with AR6 shared socioeconomic pathways relative to 1961-1990 and their associated uncertainties. Incremental increases in mean global temperature are plausible as a result of different climate scenarios at different future time horizons as shown by (solid black) circles for 1-4°C increases. Data from Fyfe et al. (2021) in IPCC (2021b).**

### 2.3.1 Rainfall Depth

There is substantial evidence that rainfall depths increase with increased global temperatures (Ali et al., 2021; Allan and Soden, 2008; Emori and Brown, 2005). In Australia, the relationship between temperature and rainfall has been investigated using both observed records (Hardwick Jones et al., 2010; Herath et al., 2018; Zhang et al., 2019) as well as modelled results (Chevuturi et al., 2018; Ju et al., 2021). Investigations into the association of rainfall with temperature in Australia have typically yielded results where the central tendencies of daily rainfall changes are in accordance with the Clausius-Clapeyron

relationship, with greater associations found for rainfalls with shorter duration and those in tropical regions (Magan et al.,





2020; Visser et al., 2021; Wasko et al., 2018). The impact of increased temperatures on rainfall depth in Australia have been found to be consistent with investigations elsewhere in the world (e.g. Allan and Soden, 2008; Gutiérrez et al., 2021).

Evidence from both observed and modelled results motivate efforts to update IDF estimates (Jayaweera et al., 2023; Schlef et al., 2023) and there has been recognition that this also applies to estimates of the probable maximum precipitation (PMP)

(Kunkel et al., 2013; Salas et al., 2020; Wasko et al., 2024a), which is the theoretical maximum precipitation for a given duration and location (WMO, 2009). To date, changes in atmospheric water vapour content, at rates that approximate the Clausius-Clapeyron relationship, have been identified as the primary driver of increased PMP estimates, while evidence of, and the ability to resolve, changes in storm efficiency have been limited (Kunkel et al. 2013). Estimates of the PMP are invariably dependent on the method and assumptions used to derive estimates and adopting the operational procedure used by

the relevant jurisdiction's authority is therefore essential to producing appropriate projections of the PMP under climate change (Stratz and Hossain, 2014).

In Australia, generalised methods are used to derive estimates of the PMP, which allow for data to be drawn from larger spatial regions to inform local estimates (see PMP zones in Fig. 1) by considering similarities in atmospheric dynamics and topography and thus the mechanisms driving extreme rainfall (WMO, 2009). Visser et al. (2022) assessed climate change impacts on PMP

estimates using the operational methods used by the Bureau of Meteorology, Australia's national weather, climate, and water agency. In their study, it was found that persisting increases in dewpoint temperatures drive increases in PMP estimates and subsequent projections of dewpoint temperatures yielded increases in PMP estimates under climate change slightly above the Clausius-Clapeyron relationship. The findings of Visser et al. (2022) are in agreement with international findings of climate change impacts on PMP estimates (Kao Shih-Chieh et al., 2019; Rastogi et al., 2017; Rouhani and Leconte, 2020).

We therefore assessed changes in storm depth based on a rate of change of 8%/°C as recommended in Wasko et al. (2024b) for critical storm durations 24 hours or longer and a rate of change of 9%/°C and 8.4%/°C for 12 hour and 18 hour storm durations respectively, noting our adopted rate of change factor is consistent with the results examining changes in rainfall depth and PMP depth with temperature in Australia. The rate of change factor was applied as follows in Eq. (2):

$$I_p = I \times \left(1 + \frac{\alpha}{100}\right)^{\Delta T} \tag{2}$$

where $I_p$ is the projected rainfall depth, $I$ is the historical design rainfall depth or intensity (e.g. from historic IDF curves or

PMP estimates), $\alpha$ is the rate of change in units of %/°C, and $\Delta T$ is the change in global (land and ocean) temperature.





### 2.3.2 Storm temporal patterns

The PMP zone-specific temporal patterns for each dam were used to estimate the baseline flood frequency (Bureau of Meteorology, 2006; Green et al., 2005; Nathan, 1992; Walland et al., 2003). These temporal patterns are comprised of around ten storm patterns for different durations and different standard catchment areas. Information on how storm temporal patterns are expected to change in a warming climate were derived from Visser et al. (2023). The changes in storm temporal patterns were examined using a measure of the proportion of the storm event duration at which 50% of the cumulative precipitation has occurred, denoted $D_{50}$, where values of $D_{50}$ range between 0 and 100%. Storms with a $D_{50}$ value of less than 50% are classified as "front-loaded events", while $D_{50}$ values greater than 50% are "rear-loaded" events. It is expected that under climate change $D_{50}$ values will slightly decrease in most regions meaning that storms are predominantly becoming more frontally loaded.

Climate change impacts on storm temporal patterns were assessed using the Köppen-Geiger zone-specific rate of change factors (%/°C) from Fig. 9 in Visser et al. (2023). These rate of change factors are shown in Table 2 for the climate zones and storm durations relevant to this study that were calculated in Visser et al. (2023). The rate of change factor for the longest duration storm was adopted when the critical duration storm exceeded the length of storms analysed in Visser et al. (2023). The calculation of the percentage change in $D_{50}$ is shown in Eq. (3).

$$\Delta D_{50} = \left[ \left( 1 + \frac{\alpha_{D50}}{100} \right)^{\Delta T} - 1 \right] \times 100 \tag{3}$$

where $\alpha_{D50}$ is the rate of change factor for $D_{50}$. The change in $D_{50}$ was calculated in response to 1°C increases in temperature and rounded to the nearest percentage change. For example, applying a temporal pattern rate of change factor of -4%/°C under a 4°C increase in global temperature would result in a $\Delta D_{50}$ of -15%.

**Table 2. Rate of change factors for storm temporal patterns by Köppen-Geiger zone (Aw: Equatorial, dry winter; Cfa: Warm temperate, fully humid, hot summer; Cfb: Warm temperate, fully humid, warm summer; Csb: Warm temperate, dry, warm summer; Cwa: Warm temperate, dry winter, hot summer; BWh: Arid, desert, hot). Numbers in parenthesis show the number of dams located in each zone.**

| Duration (hr) | Aw (1) | Cfa (6) | Cfb (7) | Csb (2) | Cwa (1) | BWh (1) |
|---|---|---|---|---|---|---|
| **12** | -0.12 | -0.58 | -0.29 | 0.17 | -0.45 | -0.50 |
| **18** | 0.03 | -0.27 | -0.90 | 0.10 | 1.07 | |
| **24** | | -0.42 | 0.26 | | | |
| **36** | | -1.09 | | | | |



For Monte Carlo simulations under historic climate conditions, the influence of natural variability in temporal patterns is accounted for by randomly selecting patterns from the available ensemble using a uniform distribution. To account for the tendency for storm patterns to become more front-loaded with warmer global temperature, the temporal patterns were sampled non-uniformly in order to achieve the targeted average shift in $D_{50}$ as shown in Eq. (4):

$$\overline{D_{50}} + \Delta D_{50} = \frac{\sum_{i=1}^{n} w_i \cdot D_{50,i}}{n} \tag{4}$$

where $n$ is the number of temporal patterns, $w_i$ is the weighting of the i[th] temporal pattern where $w_i \neq \frac{1}{n}$ when $\Delta D_{50} \neq 0$, and $D_{50,i}$ is the $D_{50}$ of the i[th] temporal pattern. In a uniform sampling of the temporal patterns $w_i = \frac{1}{n}$ and $\Delta D_{50} = 0$. The weights needed to achieve the target $\Delta D_{50}$ were determined using a random sampling of 10,000 sets of weights such that $\sum w_i = 1$, whilst minimising var($D_{50}$) to ensure as even a sampling of temporal patterns as possible to achieve the targeted shift in $D_{50}$ to an accuracy of $10^{-5}$.

### 2.3.3 Rainfall losses

Projections of changes in initial and continuing loss were undertaken, respectively, for 205 and 273 catchments across Australia (Ho et al., 2023). The catchments used in the study by Ho et al. (2023) were selected where a statistically significant relationship (at a significance level of $\alpha = 0.05$) could be found between losses and antecedent soil moisture for 3-day rainfall events that were equalled or exceeded, on average, 5 times per year (a 5 EY event). It was found that across most of Australia, rainfall losses are projected to increase under all climate change scenarios, with the largest increases seen for higher emission scenarios further into the future. Some exceptions included areas of western Tasmania and north-east Queensland where rainfall losses are projected to decrease slightly.

Projections of changes in rainfall losses were averaged over regions with similar hydroclimatic characteristics, termed "Natural Resource Management" (NRM) regions. These region-specific rainfall loss rates of change were derived from the results of Ho et al. (2023) but only used data from events that were equalled or exceeded on average once per year (1 EY), as opposed to the results presented in Ho et al. (2023), which included more common 5 EY events. The revised event selection was made here to help exclude the more frequent events where the soil moisture deficit may not have been fully satisfied by the incident rainfall. These regionally aggregated rainfall loss rates of change are documented in Wasko et al. (2024b) and are shown in Table 3 for the NRM regions relevant to the dams considered in this study. There was insufficient data to project changes in losses in the Rangelands NRM region. Consequently, values from the Monsoonal North NRM region were adopted as this was




the closest proximity NRM region to the dam located in the Rangelands NRM region. The rates of change were applied to the
260 mean parameter of the initial loss and to the constant value of the continuing loss.

**Table 3. Rates of change for rainfall losses, initial (IL) and continuing loss (CL), by Natural Resource Management (NRM) region.**

| NRM | IL (%/°C) | CL (%/°C) |
| --- | --- | --- |
| **Wet Tropics** | 0.8 | 1.4 |
| **East Coast** | 2.0 | 3.8 |
| **Central Slopes** | 1.1 | 2.0 |
| **Murray Basin** | 3.1 | 6.7 |
| **Rangelands** | - | - |
| **Monsoonal North** | 2.4 | 4.4 |
| **Southern Slopes** | 3.9 | 8.5 |
| **Southern and South-Western Flatlands** | 4.5 | 5.6 |

## 3 Results

We derived flood frequency curves in response to changes in the three different flood drivers, individually and combined, for
each case study catchment, considering increases in global temperature of 1-5°C. An example of the shifts in the derived flood
265 frequency under climate change is shown for one of the catchments in north-east Australia in Fig. 4 (the results are anonymised
here to avoid any inferences being made about the risk of overtopping to downstream communities). The red dashed horizontal
line represents the notional dam crest flood (DCF), the black curve represents the flood frequency curve under historical
climatic conditions, and progressively darker grey lines show results for increasing degrees of global warming.



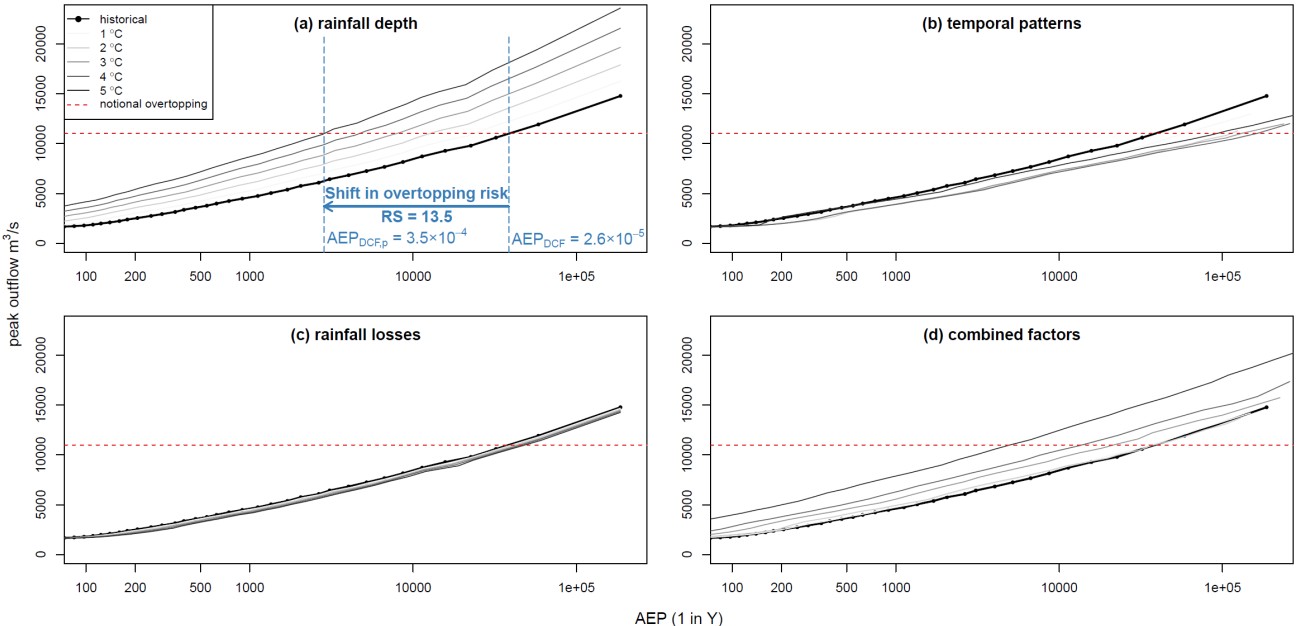

**Figure 4 Example of derived outflow flood frequency curves resulting from changes in the flood driver of (a) rainfall depth; (b) temporal patterns; (c) rainfall losses); and (d) a combination of all three flood drivers.**

The shift in the reservoir outflow flood frequency curve resulting from changes in storm depth under different degrees of global warming is shown in Fig. 4(a). The adopted rate of change value for precipitation depth in response to climate change is positive, meaning that rainfall depths are estimated to increase with increasing temperature. The derived flood frequency curves considering changes in rainfall depth under climate change are consequently steeper than the historic flood frequency curve representing an increased risk of the DCF being exceeded. The steeper flood frequency curves in response to changes in storm depth seen for this example in Fig. 4(a) are representative of the changes obtained due to increases in rainfall depths across all the case studies. In this example, the probability of exceeding the DCF is $2.6\times10^{-5}$ (approximately 1 in 38500) under historical climate conditions. This probability increases to $3.5\times10^{-4}$ under a 5°C increase in global temperatures resulting in the relative risk of overtopping increasing by 13.5.

The effect of changes in storm temporal patterns under global warming on the outflow flood frequency curve for this example case study site is shown in Fig. 4(b). In this example, the rate of change for the storm temporal patterns was negative, meaning that storms will become increasingly front loaded under climate change resulting in the probability of exceeding the DCF decreasing. However, the decreases in the probability of exceeding the DCF were not continuous with increases in global temperature. At this site, a global temperature increase of 4°C resulted in the largest change in overtopping risk. In addition to both the sign and magnitude of the temporal pattern shift the impact of changes in storm temporal patterns on flood risk are





dependent on a catchment's time of concentration and existing storm attributes. As a result, the direction of change in the probability of extreme floods resulting from changes in storm pattern are specific to each catchment.

Shifts in the dam overtopping risk in response to changes in rainfall loss under climate change are shown in Fig. 4(c). For this
site, the increases in rainfall losses are small relative to the design rainfall depths at the probabilities of interest to the DCF resulting in a very small decrease in the risk of floods exceeding the DCF under climate change. The impact of changes in rainfall losses on the risk of exceeding the DCF differed in magnitude between catchments in different regions. As seen in Table 3, the rates of change range from 0.8-4.5%/°C for initial losses and 1.4-8.5%/°C for continuing losses and the resulting decreases to the risk of exceeding the DCF were notable for some locations.

The combined impacts of changes in storm depth, storm temporal pattern, and rainfall loss in response to 1-5°C of global warming are shown for this example case study site in Fig. 4(d) showing an overall increase in the risk of floods exceeding the DCF. A comparison of Fig. 4(a)-(c) with Fig. 4(d) shows that changes in rainfall depth exert the largest influence over changes in flood risk under climate change, a finding that was universal across the dams investigated. From these derived flood frequency estimates we calculated the relative shift in the risk of the DCF (see Eq. (1)) to summarise the results recalling that
values less than 1 represent a decreased risk while values greater than 1 represent an increased risk in floods exceeding the DCF.

The shifts in overtopping risk due to each flood driver are shown as box plots in Fig. 5(a) – (c) with Fig. 5 (d) showing the response to the combined impacts of all three flood drivers. Each box plot is a summary of the results across the 18 dams. For each flood driver, five box plots are shown representing increasing degrees of global warming. Similar to the results shown in
Fig. 4, a comparison between Fig. 5(a) and (d) show that the flood risk resulting from changes in all the flood drivers is most influenced by changes in the rainfall depth. However, the differences between Fig. 5(a) and (d) also reveal that the effects of changes in temporal patterns and rainfall losses are not negligible, despite their relatively small impacts when considered individually (see Fig. 5(b) and (c) noting the different scales on the y-axes compared to Fig. 5(a) and (d)). Interestingly, while the impacts of rainfall losses on reducing flood risk are magnified with increased global warming, the impact of changes in
temporal patterns do not necessarily change uni-directionally with increased global temperature. In addition, the direction of change in the storm temporal pattern was not indicative of the direction of change in the derived flood frequency curve. All catchments show decreases in flood risk in response to 1-3 °C global temperature increases, while some of the catchments experience an increased flood risk under 4-5°C of warming. These results indicate that changes in peak outflows in response to changes in temporal patterns are catchment specific and likely dependent on the catchment's time of concentration and other
storm attributes such as the spatial distribution of rainfall.



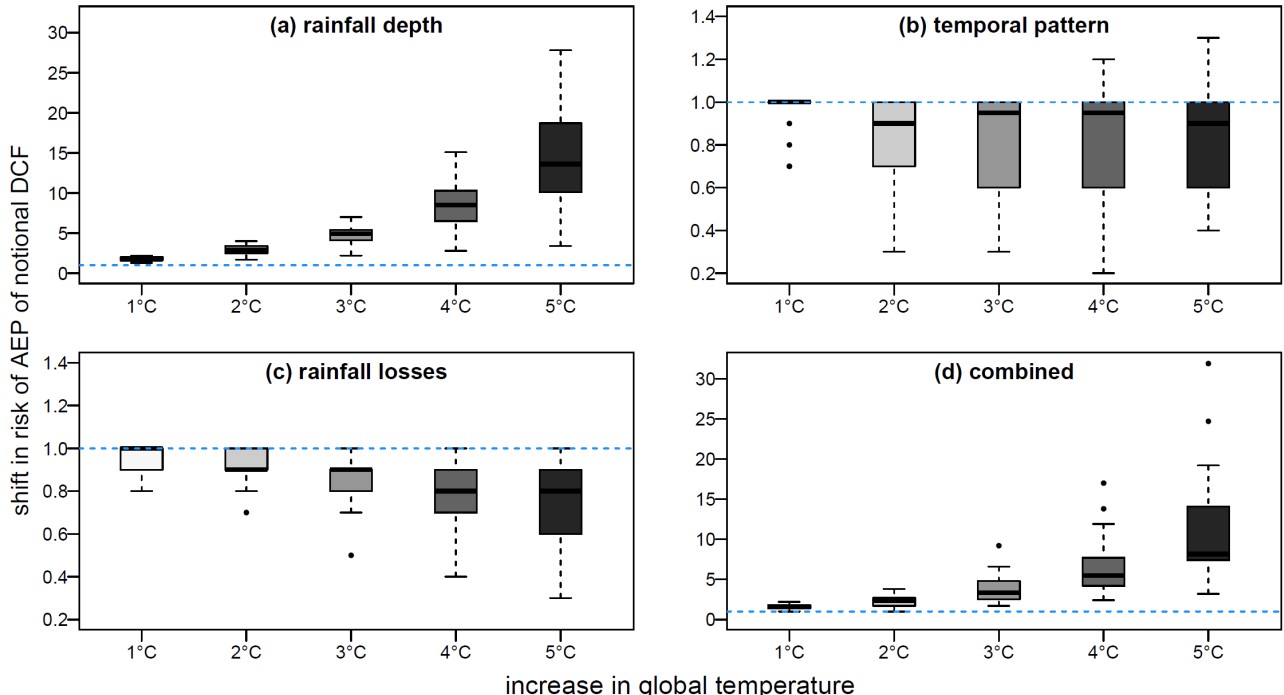

**Figure 5. Box plots of the relative shift in the annual exceedance probability (AEP) of the dam crest flood (DCF) for all 18 dams resulting from changes in (a) rainfall depth (outliers not shown), (b) temporal pattern, (c) rainfall losses, and (d) all three flood drivers combined. Box plots show the median and interquartile range (IQR). Whiskers show the minimum and maximum values that lie within 1.5 times the IQR of the median value. Outliers are not shown. Dashed blue line represents the historical baseline.**


Overall, the impact of global warming increases the exceedance probability of the DCF at most locations, noting that a temperature increase of 1ºC approximates present conditions due to the baseline period of 1961-1990 adopted here (see Table B 1 showing projections of global mean surface temperature changes for the current and near term period). The degree to

which climate change is projected to increase the probability of a flood exceeding the DCF appear to be catchment- and dam-dependent. For example, Fig. 5(d) shows that under a 4ºC increase in global temperature, which approximates projected temperatures towards the end of the 21st century under medium to high emission scenarios (see Fig. 3 and Table B 1), the risk of overtopping due to all drivers combined ranges from 2.4 to 17 times (median of 5.5) greater than the historic risk across the different dams.



## 4 Discussion

It is necessary to translate knowledge from the realms of scientific theory, investigation, and experimentation to practice (Kiem and Austin, 2013). The estimation of flood risk under climate change is a challenging endeavour due to the dynamic nature of flood mechanisms even under stationary climate assumptions. Yet projecting flood risk under climate change is critical to ensuring that the risk is adequately managed particularly in high hazard systems with long (i.e. several decades to century-long) economic life-spans, such as large dams and nuclear power plants, which need to withstand extreme flood events under both current and future climate conditions. A scientific review aimed at consolidating the available information relevant to estimating floods under climate change in Australia was recently conducted by Wasko et al. (2024a) and the findings from the review were used to inform the rates of change used in this study. The work presented here was undertaken contemporaneously to an update of the Australian flood guidelines (Wasko et al., 2024b) also based on the summary findings of Wasko et al. (2024a) while another study focused on more frequent events by O'Shea et al. (2024). The exchange between developing the new guidelines and the study presented here and that of O'Shea et al. (2024) was intended to ensure the practicality of the new guidelines, demonstrate an approach that could be translated globally for estimating flood risk under climate change elsewhere in the world, as well as providing insights into the impact of climate change on flood risk for the case studies.

The 18 case studies considered here represent a broad sample of catchment and dam sizes, with all dams meeting the ICOLD classification of large dams (ICOLD, 2011), located across a range of climate zones. We found that changes to the overtopping risk were most sensitive to changes in precipitation depth in response to global warming across all dams. While changes in temporal patterns and rainfall losses had a relatively smaller impact, they were not inconsequential. For many of the smaller catchments located in the southern temperate regions of Australia changes in rainfall losses and temporal patterns moderated the increased flood risk resulting from increased rainfall depth under climate change. It is also plausible that shifts in temporal patterns and rainfall losses will be more critical for more frequent floods compared with the extreme floods considered here, where changes in the extreme rainfalls relevant to overtopping floods overwhelm the changes in rainfall losses or temporal patterns. A supporting result was found in a study by O'Shea et al. (2024) focused on frequent (i.e. 1 in 5 AEP) and rare (i.e. 1 in 50 AEP) floods in response to climate change impacts on rainfall depth and rainfall losses. They found that flood peaks were more sensitive to climate change for more frequent floods compared with rare floods. O'Shea et al. (2024) also found a heightened sensitivity in the catchment located in a temperate climate compared to the catchment located in a tropical region. The increased sensitivity in response to shifts in rainfall losses can be attributed to the smaller runoff ratios associated with more frequent flood events as well as the smaller runoff ratios typical of Australian catchments located in temperate climates (Wasko and Guo, 2022), which make them more sensitive to changes in rainfall losses.

The ability to generalise likely climate change impacts on relative changes in overtopping risk based on dam-specific attributes, such as climate zone, catchment size, rainfall-runoff characteristics, reservoir capacity, and the configuration of outlet works





(e.g. whether dam outflows are controlled by gates or fixed crest spillway) would be a valuable finding. Thus far, the sample size of catchments investigated both here and in the study by O'Shea et al. (2024) are insufficient to definitively conclude whether sensitivities in flood risk under climate change can be associated with specific catchment characteristics, or, for this study, specific dam characteristics. At present, projecting estimates of dam overtopping flood risk as an indicator of dam failure requires a thorough site-specific investigation.

We used a conservative assumption that reservoirs are at full supply level prior to the storm, an assumption that provides a worst-case scenario with respect to the subsequent estimation of outflow flood risk. It is anticipated that, all other things being equal, climate change will result in larger demands for water resulting in increased drawdown and evaporation of reservoirs prior to floods thereby increasing the potential for dams to attenuate floods, while impacts on the frequency and depth of storms will modify reservoir recharge. Indeed, the study by Lompi et al. (2023) for a dam in northern Spain showed that the near-term (2040) overtopping risk was reduced under a moderate emission scenario as a result of increased reservoir airspace. The interaction between increased reservoir airspace prior to a storm and increased precipitation depth under climate change has yet to be explored in the context of examining overtopping risk for the dams in this study. Such an investigation would require estimation of a distribution of the initial reservoir level under climate change for each dam. Continuous flood models are well suited to assessing higher frequency flood events relevant to changes in water demand and supply, in contrast to event-based models used here to estimate extreme flood risk relevant to the economic design life span of dams. However, the difference in time scales relevant to estimating changes in reservoir airspace (i.e. years to decades) means that the consideration of decadal variability, in addition to climate change, becomes pertinent (Kiem et al., 2003; Malakar et al., 2024; Micevski et al., 2006). Information from decadal climate forecasts could potentially be used to inform shifts in stochastic weather generators relevant to continuous flood modelling (Dykman et al., 2024; Steinschneider and Brown, 2013) or rainfall intensity frequency duration curves relevant to event-based modelling (Jayaweera et al., 2023). In addition to accounting for climate variability and change, a comprehensive model of reservoir airspace would also consider broad policy decisions regarding the augmentation of water supply, demand management, and population change, considerations that are in the realm of deep uncertainty.

The change in global temperature was used as the covariate for projecting impacts of climate change on the flood drivers considered in this study, an approach recommended by Kunkel et al. (2020). General circulation models are able to model temperature with a high degree of confidence at both global and regional scales (IPCC, 2021a). Our approach therefore capitalises on one of the more reliable outputs from modelling projections of climate change. The choice of using a global spatial scale was made to be both consistent with IPCC projections as well as being representative of the primary driver of changes in atmospheric circulation and moisture stores. While it may seem intuitive to employ regional, or more local downscaled, projections of temperature given their demonstrated fidelity in general circulation models, the use of temperature on smaller spatial scales as a covariate of extreme rainfall has yielded inconsistent results (Chan et al., 2016; Wang et al.,





2017). In contrast, global temperatures have been found to be better predictors of Australian rainfall (Jayaweera et al., 2024; Roderick et al., 2020). Furthermore, conditioning our assessment of changes in flood risk under climate change on global temperature mean that the flood assessment can be conducted independent of projections of climate change, which is an

involved process that includes developing scenarios, evaluating the suitability of general circulation models, evaluating downscaling and bias correcting methods, and selecting a manageable and representative suite of ensemble runs to consider. There is, however, value in finer spatial resolution numerical models in assessing flood risk under climate change for new dam sitings as local climate impacts resulting from changes in land use and land cover have been shown to result in increases in estimations of the PMP of over 10% (Stratz and Hossain, 2014).

The work presented here provides a basis for estimating changes in overtopping risk resulting from changes in the salient flood drivers. Estimating the risk of overtopping floods under climate change can be used to inform broader assessments of compounding dam risk that include consideration of rates of sedimentation and changes in the exposure of downstream populations and industries reliant on reservoir storages over time (Fluixá-Sanmartín et al., 2018). Such estimates can also inform decisions regarding relicensing or reoperating existing dams under climate change (Ho et al., 2017; Pittock and

Hartmann, 2011; Watts et al., 2011). We note that our study only used the approximate central tendencies of the rates of change for adjusting rainfall depths, temporal patterns, and rainfall losses and only presented the best estimate of overtopping flood risk under climate change. We recommend that future studies explore the relative sources of uncertainties associated with the rates of change used for considering climate change and existing aleatory and epistemic uncertainties.

In addition to developing sound guidelines for practitioners to implement, the challenge of communicating and improving the

understanding of flood risk in the broader population remains (Pielke, 1999) and with it the need to improve the communication of flood risk (Read and Vogel, 2015). Updating estimates of flood risk under climate change can also ensure ex post evaluations of flood disasters are appropriately informed and attributions to climate change are not erroneously made at the expense of identifying and resolving other factors (Doss-Gollin James et al., 2020). It is crucial that communications of flood risk occur in parallel with improving understanding of the intended utility of water infrastructure so that levels of public confidence and

expectations with respect to the preventative capacity of flood infrastructure are reasonable (Lave and Lave, 1991). We demonstrate here that it is possible to estimate changes in extreme flood risk under climate change, but there is a societal imperative to act upon this knowledge and to recognise our increasing exposure to flood risk that results in part from climate change but more broadly from an expanding portfolio of assets in flood zones (Kundzewicz et al., 2014) that can be motivated by perverse economic incentives (Gourevitch et al., 2023).





## 5    Conclusion


We present the first assessment of changes in flood-induced dam overtopping risk under climate change based on contemporary understandings of climate change impacts on key flood drivers. Our assessment explicitly considers climate change impacts on rainfall depth, storm temporal patterns, and rainfall losses and was conducted in a manner designed to be readily adopted in industry applications. We estimated projections of flood risk conditional upon scenarios of increased mean global

temperatures using event-based flood modelling and Monte Carlo simulation to consider the joint probabilities of the salient flood drivers.

For the 18 dams examined, we found that the impacts of climate change under 4ºC global warming increases the risk of floods exceeding the dam crest flood by 2.4-17 times (with a median value of 5.5) compared to estimates based on historic climate conditions. Furthermore, current levels of global warming relative to the period used to inform historic flood risk estimates in

Australia mean that the risk of floods exceeding the dam crest flood is already more than twice as probable for four of the 18 dams investigated. Of the three flood drivers considered, changes in extremely rare rainfall depths relevant to dam crest floods had the largest impact increasing the overtopping flood risk by around an order of magnitude for most dams under 4ºC of global warming. In contrast, the change in extreme flood risk resulting from changes in temporal patterns were marginal and the magnitude of impacts appear contingent on how runoff is routed through the catchment. Changes in rainfall losses slightly

decreased the risk of overtopping floods across all locations resulting in the impact of increased rainfall intensity being slightly dampened.

Given the complex interaction of flood drivers, it is currently not possible to provide heuristics for estimating changes in flood risk under climate change based on attributes such as catchment location, climate zone, or catchment or dam size. Assessments of climate change impacts on flood risk need to instead, at present, be assessed in a site-specific manner. Our study provides

a practical approach for estimating extreme flood and dam overtopping risk under climate change that aligns with approaches widely used by practitioners making it feasible to be adopted globally.

### Appendix A

For each case study dam, a semi-distributed conceptual node-link model of the catchment was used to represent the storage and routing of streamflow, where nodes represent either the centroid of a subarea where rainfall is added or junctions in the

conceptual stream network, and links represent the main tributary streams along which streamflow is routed. Such node-link networks provide a simplified characterisation of the drainage network and are commonly used in event-based modelling (Pilgrim and Cordery, 1993). The catchments were divided into sub-areas where rainfall was assumed to be uniform within each subarea.





The initial loss continuing loss model (ILCL) was used to partition rainfall into rainfall losses and runoff that is then routed

through the catchment stream network. Rainfall losses are separated into two components being the initial loss, which represents the depth of rainfall required to sufficiently wet the catchment before runoff commences, and the continuing loss, which is the rate of rainfall loss that occurs once the initial loss has been satisfied through to the end of the rainfall event. The runoff, or rainfall excess, $X_t$, at each time step is expressed as shown in Eq. (A1):

$$X_t = \begin{cases} 0 & for \left( \sum_{i=1}^{t} P_i \right) \leq IL \\ max(0, P_t - CL) & for \left( \sum_{i=1}^{t} P_i \right) > IL \end{cases} \tag{A1}$$

Where $P$ is the rainfall depth (mm) and the subscript $t$ or $i$ denotes the timestep (hr), $IL$ is the initial loss parameter, and $CL$ is

the continuing loss parameter (mm/hr). The ILCL model  was selected from a range of commonly used rainfall loss models as it is recommended design flood estimation in Australia (Hill and Thomson, 2019) and it has been shown to be most suitable for applications where estimates are made for floods that exceed the magnitude of observations (O'Shea et al., 2021).

The rainfall excess was then routed through the catchment stream network using a non-linear storage routing power function based on continuity as shown in Eq. (A2). This was used to model the attenuation and delay of runoff (i.e. overland flow) from

a subarea, the routing of a hydrograph through a reach, as well as the routing of a hydrograph through a reservoir.

$$S = kQ^m \tag{A2}$$

where $S$ is the storage (m$^3$), $Q$ is the outflow (m$^3$/s), $m$ is a dimensionless exponent, and $k$ is a dimensional empirical coefficient. A value of m = 0.8 widely adopted in general practice was used here and represents the degree of non-linearity in the catchment response. The coefficient, $k$, is the product of $k_c$, which represents the relative storage and delay of streamflow of the catchment, and $k_r$, which is the relative delay time of each reach storage: $k_c$ was obtained by calibration to observed flood events, while $k_r$

is dependent on the relative reach length. Baseflows were added to the reservoir inflows to account for delayed streamflow contributions from prior rainfalls (these were generally less than 1% of the peak flows), and outflows from the dam were calculated using appropriate storage-outflow relationships representative of the dam storage configuration and outlet works.

The outflow flood frequency curve for each dam was derived in response to the critical duration storm using the Total Probability Theorem (Haan, 2002; Nathan et al., 2003; Nathan and Weinmann, 2019b). Rainfall depths were sampled by using

a stratified Monte Carlo sampling over the standardised normal probability domain of rainfall depths. This stratified sampling





enables rare rainfall events to be sufficiently sampled. Here, the rainfall distribution was stratified into 100 divisions with 200 samples in each division, thus each flood frequency curve was based on the simulation of 20,000 flood events.

The flood exceedance quantiles were then calculated using Total Probability Theorem as shown in Eq.(A3).

$$P(X > x) = \sum_i P(X > x|C_i)p(C_i) \tag{A3}$$

where $C_i$ is the conditioning variable (i.e. rainfall) with values that fall within the $i^{th}$ interval, $X$ is the calculated flood value
and hence the term $P(X>x|C_i)$ is the conditional probability that the flood outcome $X$ generated from $C_i$ exceeds $x$. The term $p(C_i)$ is the probability that the conditioning variable falls within the $i^{th}$ interval.

In addition to the sampling of rainfall depths, temporal patterns were stochastically selected from an ensemble obtained from observed storms (Green et al., 2019), and initial rainfall losses were sampled from an empirical distribution based on the findings of Hill et al. (2014).

**Appendix B**

**Table B 1. IPCC Sixth Assessment Report (AR6) global mean surface temperature change projections for four Shared Socioeconomic Pathway (SSP) climate scenarios relative to 1961-1990 baseline (which is notionally representative of the mid-point for much of the information used to derive the design information provided in ARR2019). The 90% uncertainty interval is provided in parentheses†.**

| Time horizon | SSP1-2.6 | SSP2-4.5 | SSP3-7.0 | SSP5-8.5 |
|---|---|---|---|---|
| **Current and near-term (2021-2040) (○C)** | 1.2 (0.9-1.5) | 1.2 (0.9-1.5) | 1.2 (0.9-1.5) | 1.3 (1.0-1.6) |
| **Medium-term (2041-2060) (○C)** | 1.4 (1.0-1.9) | 1.7 (1.3-2.2) | 1.8 (1.4-2.3) | 2.1 (1.6-2.7) |
| **Long-term (2081-2100) (○C)** | 1.5 (1.0-2.1) | 2.4 (1.8-3.2) | 3.3 (2.5-4.3) | 4.1 (3.0-5.4) |

*†Projections are adapted from the Summary for Policymakers of the Working Group I Contribution to the Intergovermental Panel on Climate Change Sixth Assessment Report (Fyfe et al., 2021; IPCC, 2021b)*

**6   Author contribution**

Michelle Ho: conceptualisation, methodology, formal analysis, data curation, visualization; Declan O'Shea: conceptualisation, methodology, data curation; Conrad Wasko: conceptualisation, methodology, data curation; Rory Nathan: conceptualisation,



methodology, formal analysis, software, resources, funding acquisition, project administration; Ashish Sharma: conceptualization, resources, funding acquisition, project administration. All authors contributed to the writing of the original draft, review, and editing.

## 7 Competing interests

The authors declare that they have no conflict of interest.

## 8 Acknowledgements


The HRS streamflow data set is publicly available from www.bom.gov.au/hrs. Sub-daily rainfall data is available for a cost from the Australian Bureau of Meteorology or from the authors at reasonable request. Thanks to Matthew Scorah for processing the data for the temporal patterns. Catchment models used for verifying R$^2$ORB were made available by the industry partners (listed here under "industry support"). This project was supported the Australian Research Council (ARC) Discovery Projects
DP200101326 and industry support from the Queensland Department of Natural Resources, Mines and Energy, Hydro Tasmania, Melbourne Water Corporation, Murray-Darling Basin Authority, Seqwater, Snowy Hydro, Sunwater, West Australia Water Corporation, and WaterNSW. Declan O'Shea acknowledges the support of an Australian Government Research Training Program Scholarship and the University of Melbourne Lochrie Engineering Scholarship. Conrad Wasko acknowledges funding from the Australian Research Council (ARC) DE210100479 and the University of Sydney Horizon
Fellowship.

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
