# Peer review of "The impact of climate change on dam overtopping floods in Australia"

_Hydrology and Earth System Sciences, 2024_

## Author Response (AR1)

**Referee 1**

**General comments**

This paper addresses a critical and underexplored issue—the implications of climate change on dam hydrological safety. The topic is highly relevant given the increasing frequency and severity of extreme weather events, making the findings applicable to dam owners, policymakers, and climate adaptation professionals globally. The study offers a robust method for addressing uncertainties, and its analysis of 18 large dams across varied Australian climate regions is particularly noteworthy.

However, while the study's objectives are clear and the methodology is sound, there are areas for improvement:

1. First of all, the title of the article clearly refers to the impact of climate change on dam overtopping. However, most of the paper focuses on the impact on the hydrological loads to dams, neglecting the specific aspects related to dam safety. Furthermore, the only considerations regarding dams are taken into account in a very simplified manner (e.g., the assumption of the reservoirs being at a full supply level prior to the storms). The article title does not embody the methodology and results presented, and cannot be accepted as a valid title.

To provide a more precise description of the study, we have replaced the title of "The impact of climate change on dam overtopping flood risk" with "The impact of climate change on dam overtopping floods". Please refer to our response in point 2 where we provide further explanation.

2. Throughout the entire manuscript, the term "risk" is misused (even in the title). In the dam safety context (and in any context related to natural hazards), the risk is defined as the combination of a potential hazard and its consequences. However, in the manuscript only the occurrence of the hazard and its probability are studied and quantified. Therefore, the manuscript should be reviewed and the term "risk" should be adequately replaced by "probability" or "hazard", depending on the case.

The estimation of flood exceedance probabilities is the basis of assessing hydrological risk-based designs in engineering and we had been using the term "risk" in this context. However, we did not clarify this context and we recognise that this application of the term is used variously in the literature. We have therefore replaced this terminology throughout the manuscript, as well as the title, as suggested to provide a more precise description of our analysis by adopting the reviewer's suggested terminology of "probability".

3. The methodology is quite dependent on the Australian context. The introduction and discussion sections could better frame the global implications of the findings, as they currently focus primarily on Australia. Specify what assumptions and methods can be replicated to other regions or contexts. In line 342, the authors say that the approach "could be translated globally for estimating flood risk under climate change elsewhere in the world", which is not clear.

We thank the reviewer for pointing this out. We have added text to introduce the global context in the introduction (see also our response to reviewer 2 comments #4 and #5). Providing specifics with respect to model and data requirements will make these methods globally applicable and this has been included in the third paragraph of the discussion.

4. The graphical abstract includes a graph representing the dam crest flood level vs the exceedance probability:

**a. This graph is not represented in the article.**

**b. The y-axis does not correspond to the exceedance probability.**

The reviewer is correct – the graphical abstract is not from the article. It was specially designed for the purpose of providing an overview of the research and general results rather than to convey specific results and we believe that it serves this purpose.

The reviewer is also correct that the y-axis does not correspond to the exceedance probability – the y-axis represents the reservoir level and this was communicated using an icon of a dam with water levels along the y-axis. To clarify this, we have added a text label to the y-axis in addition to the icon and relocated the x-axis label to improve clarity.

**Specific comments**

**Introduction**

5. The introduction provides a strong rationale for the study. However, it lacks a succinct statement of the research gap. Explicitly contrast the current study with previous works on flood risk and dam safety.

We have revised the last paragraph of the introduction and believe that we have now provided an explicit contrast between our study and previous work in the last two paragraphs of the introduction. We relocated details of the analysis and data from the final paragraph to Section 2 (Materials and methods) and rearranged the last paragraph of the introduction to clarify the research statement. We believe that addressing the reviewer's recommendation of using precise terminology with respect to the word "risk" has also improved the clarity of our statement explaining the research gap.

6. The global context is underdeveloped. Adding examples from other regions (e.g., Europe or Asia) could broaden the impact.

The two studies we reference that quantified changes in the probability of dam overtopping floods were located in Taiwan and Spain and a reference to these locations has been added. We state that there are few other studies that specifically investigate the changes in dam outflow flood frequencies under climate change. However, we have included additional references to previous studies that investigate dam overtopping floods (see our response to Referee 2, comment #5) and these studies are demonstrated for dams located in Taiwan, Italy, Korea, The United States, and China.

**Materials and methods**

7. Line 121: the description of the R2ORB emulator needs more detail. A brief explanation of how it works and its advantages compared to other tools would be helpful.

We have included more detail on the R2ORB emulator in Section 2.2. when R2ORB is first mentioned. We have revised the text to explain that R2ORB uses data inputs and performs the calculations outlined in Fig 2, which shows a schematic of the event-based modelling process. We have added that by using an emulator of RORB, we were able to focus our calculations on the aspects of flood hydrology modelling that are most relevant in the exploration of climate change impacts on dam hydrology, namely the catchment runoff-routing and reservoir routing to estimate peak reservoir outflows. We also included a description of our approach to calibrating and validating the R2ORB models in Appendix A. We have also added that R2ORB follows the basic generic modelling structure of event-based

conceptual rainfall runoff models, namely, partitioning of rainfall into losses and excess and attenuation of the flood through channels and reservoirs. There are many hydrological tools that can model runoff responses to rainfall at an event timescale and we do not think that a listing and comparison to these tools would be of material value to this paper.

- 8. Line 129: "we assumed that the reservoir was at a full supply level prior to the storm": this is a strong simplification of the methodology. Please justify:
  - a. The reasons (lack of data, lack of time...)
  - b. The potential impact (have you done an example calculation?)

In terms of dam safety, this is equivalent to not considering antecedent catchment wetness when calculating floods. I strongly recommend the authors to at least perform one example analysis of this effect. This is important in an article that focuses on dam safety (it's in the title).

We include in our justification for assuming a full supply level in all the dams that it provides a worst-case scenario for estimating the probability of a dam crest flood. We have also added that the modelling of reservoir levels considering climate change impacts had only been conducted for two of the dams and these were based on specific future scenarios making them incompatible with our analysis approach, which is based on changes in global temperature. While it is possible to include initial reservoir level in a Monte Carlo analysis, the assumptions underlying the shifts in the marginal distribution are subject to deep uncertainty due to the future changes in operating conditions that are required, which in our experience have a much greater influence on reservoir levels than the change in antecedent conditions. We have provided some additional discussion on this point in Section 2.2 and in the third paragraph of the discussion.

9. Line 134: "outflow hydrograph": do you mean the catchment's outflow hydrograph, or the reservoir's outflow hydrograph? Please harmonize the vocabulary throughout the paper.

At the first occurrence in *Section 2.2 Event-based modelling*, we have clarified that the "inflow hydrograph" refers to the reservoir inflow hydrograph while the "outflow hydrograph" refers to the reservoir outflow hydrograph.

10. More information on the Monte Carlo simulation framework is needed in Appendix A and in the description of the methodology.

We have added details of the Monte Carlo simulation to the methods under section 2.2. Event-based modelling. We state the variables that are stochastically sampled and the number of flood events that are simulated to derive a flood frequency curve. The reader is then directed to Appendix A where the last three paragraphs specify how each flood factor is sampled. We have added an additional detail of the distribution used to stochastically sample the storm temporal patterns.

**11. Line 162: impacts of climate change on what?**

This has been revised to read: "The impacts of climate change on the overtopping probability ..."

12. No mention to dam operations is presented here. What are the assumptions? How dams are considered operated in the baseline period and in the future periods? Just a short indication is given in line 138.

We have added that dam operations in response to an overtopping flood under climate change are assumed to remain the same as historical operations. We have noted that the operational impacts are encompassed in the relationships between reservoir height and outflow provided by the dam owners. This point is also relevant to our response to item 8) above.

**Results**

13. The results are presented effectively, but their practical implications could be expanded to aid decision-makers.

We note that in section 4, we discuss that assessments of dam overtopping floods under climate change "can be used to inform broader assessments of compounding dam risk that include consideration of rates of sedimentation and changes in the exposure of downstream populations and industries reliant on reservoir storages over time.....(and has relevance to) decisions regarding relicensing or reoperating existing dams under climate change..."
We have provided the reviewer with more granular detail of the results (see our response to comment #18), however, we feel that our presentation of the results in Figure 5 is more suitable for a journal paper.

14. Line 263: specify the three flood drivers.

Thank you. This has been addressed.

15. Line 272: "the reservoir outflow flood frequency curve" should be "the reservoir inflow flood frequency curve" because it refers to the DCF, which is the flood entering the dam's reservoir.

The dam crest flood is dependent on the outflow rate from the dam and we are in fact showing the reservoir outflow flood frequency curves in Fig 4. We have included that the outflow rate corresponding to dam crest reservoir levels is indicated in these figures.

16. The RS factor does not illustrate the importance of the underlying AEP (i.e., how unsafe is a dam under historical conditions). When presented, results should simultaneously show the base AEP and the RS factor. Moreover, given that the results are anonymized and Figure 5 is just a summary of the results for the 18 dams analyzed, maybe an anonymized figure (similar to Figure 4, but only for the combined effect) in an Appendix could help shedding light on this issue.

We show a summary of the outflow frequency curves across all dams in response to all three flood factors combined with the y-axis standardised by the dam crest flood level in appendix C. We have also included the range of historical dam crest flood AEPs across the 18 dams under section 2.3 to help justify our use of a metric representing the relative shift in AEP under climate change. We do not think it would be helpful or relevant to provide comment on how unsafe the dams are under historical conditions: apart from the fact that the owners would not want to see this point highlighted, the assessment of "dam safety" is a complex task that needs to take into account dam-specific engineering factors that lie outside the scope of this paper.

17. Synthesize somewhere the simplifications assumed in the methodology and the potential improvements.

We have clarified that we have included the key aspects of modelling a catchment's rainfall-runoff relationship and dam operations relevant to assessing climate change impacts on dam crest flood frequency and have expanded on the second to last paragraph of our discussion where we've outlined our simplifications. We also included here that dam owners would be

able to implement our analysis approaches for assessing climate change impacts on changes in the likelihood of dam crest floods using their models that could include more detailed representations of the catchment, dam operations, and initial reservoir levels.

18. The Results section is somehow succinct and could benefit from more detailed analysis. For instance, the authors haven't studied in more detail the hydrological response of each catchment to the changes in the rainfall parameters (rainfall losses, storm temporal patterns...). Figure 5 is presented as a black box without details and hardly exploitable.

We have derived detailed hydrologic results for each dam, and Figure 5 is intended to provide a summary that shows the variation in outcomes due to differences in dam and catchment configuration. We have anonymised the results for all the dams and have included these results in the four tables below for each flood factor and for the combined impacts of these factors. We have also included plots of flood frequency curves for all the dams under the combined factors under climate change in a new Appendix C. We believe that the summary shown in Figure 5 is a more suitable presentation of our results.

Table 1: Summary of risk multipliers for the probability of the notional DCF under precipitation scaling

| Dam number | 1°C | 2°C | 3°C  | 4°C  | 5°C  |
|------------|-----|-----|------|------|------|
| 1          | 2.7 | 7.0 | 16.9 | 38.8 | 94.3 |
| 2          | 2.3 | 5.4 | 12.3 | 29.3 | 59.1 |
| 3          | 1.7 | 2.8 | 5.0  | 9.0  | 15.2 |
| 4          | 1.8 | 3.3 | 5.4  | 9.1  | 14.1 |
| 5          | 1.9 | 2.5 | 4.1  | 6.6  | 9.3  |
| 6          | 2.1 | 4.0 | 7.0  | 11.9 | 18.7 |
| 7          | 2.5 | 4.8 | 7.8  | 17.6 | 40.7 |
| 8          | 1.6 | 3.0 | 4.8  | 8.0  | 13.1 |
| 9          | 1.6 | 3.1 | 4.9  | 7.2  | 12.0 |
| 10         | 1.3 | 1.7 | 2.2  | 2.8  | 3.4  |
| 11         | 1.9 | 3.4 | 6.1  | 10.3 | 18.8 |
| 12         | 1.7 | 2.7 | 4.1  | 6.2  | 8.4  |
| 13         | 2.1 | 2.7 | 5.3  | 10.2 | 19.6 |
| 14         | 1.6 | 2.7 | 4.6  | 7.4  | 11.9 |
| 15         | 1.9 | 3.4 | 5.2  | 9.1  | 14.6 |
| 16         | 2.0 | 4.0 | 8.1  | 15.1 | 27.8 |
| 17         | 1.4 | 2.4 | 3.1  | 5.2  | 10.5 |
| 18         | 1.3 | 1.9 | 2.8  | 3.7  | 5.4  |

Table 2: Summary of risk multipliers for the probability of the notional DCF under temporal pattern scaling

| Dam number | 1°C | 2°C | 3°C | 4°C | 5°C |
|------------|-----|-----|-----|-----|-----|
| 1          | 1.0 | 0.8 | 1.0 | 1.0 | 0.9 |
| 2          | 1.0 | 0.6 | 0.6 | 0.6 | 0.6 |
| 3          | 1.0 | 1.0 | 1.0 | 1.0 | 1.0 |
| 4          | 0.8 | 0.8 | 0.8 | 0.8 | 0.9 |
| 5          | 1.0 | 1.0 | 1.0 | 1.0 | 1.0 |
| 6          | 1.0 | 1.0 | 1.0 | 1.0 | 1.0 |
| 7          | 1.0 | 1.0 | 1.0 | 1.0 | 1.0 |
| 8          | 0.7 | 0.3 | 0.3 | 0.2 | 0.4 |
| 9          | 0.7 | 0.8 | 0.6 | 0.6 | 0.6 |
| 10         | 0.9 | 0.9 | 0.9 | 0.9 | 0.8 |
| 11         | 1.0 | 0.9 | 0.9 | 0.9 | 0.9 |
| 12         | 1.0 | 1.0 | 1.0 | 1.0 | 1.0 |
| 13         | 1.0 | 0.6 | 0.6 | 0.6 | 0.6 |
| 14         | 1.0 | 0.7 | 0.7 | 0.7 | 0.7 |
| 15         | 1.0 | 0.6 | 0.6 | 0.6 | 0.6 |
| 16         | 1.0 | 1.0 | 1.0 | 1.2 | 1.2 |
| 17         | 1.0 | 1.0 | 1.0 | 1.0 | 1.0 |
| 18         | 1.0 | 1.0 | 1.0 | 1.0 | 1.3 |

Table 3: Summary of risk multipliers for the probability of the notional DCF under rainfall loss scaling

| Dam number | 1°C | 2°C | 3°C | 4°C | 5°C |
|------------|-----|-----|-----|-----|-----|
| 1          | 0.8 | 0.7 | 0.5 | 0.4 | 0.3 |
| 2          | 0.9 | 0.9 | 0.9 | 0.8 | 0.8 |
| 3          | 1.0 | 0.9 | 0.9 | 0.8 | 0.7 |
| 4          | 1.0 | 1.0 | 0.9 | 0.9 | 0.9 |
| 5          | 1.0 | 1.0 | 1.0 | 1.0 | 1.0 |
| 6          | 1.0 | 1.0 | 1.0 | 1.0 | 1.0 |
| 7          | 1.0 | 1.0 | 1.0 | 1.0 | 0.9 |
| 8          | 1.0 | 0.9 | 0.9 | 0.8 | 0.8 |
| 9          | 0.9 | 0.9 | 0.8 | 0.7 | 0.6 |
| 10         | 1.0 | 1.0 | 1.0 | 1.0 | 1.0 |
| 11         | 1.0 | 1.0 | 0.9 | 0.9 | 0.9 |
| 12         | 1.0 | 1.0 | 0.9 | 0.9 | 0.9 |
| 13         | 1.0 | 0.8 | 0.7 | 0.6 | 0.5 |
| 14         | 0.9 | 0.8 | 0.7 | 0.6 | 0.5 |
| 15         | 0.9 | 0.9 | 0.8 | 0.8 | 0.7 |
| 16         | 1.0 | 0.9 | 0.9 | 0.8 | 0.8 |
| 17         | 1.0 | 1.0 | 0.9 | 0.7 | 0.7 |
| 18         | 0.9 | 0.8 | 0.8 | 0.7 | 0.6 |

Table 4: Summary of risk multipliers for the probability of the notional DCF under combined impacts

| Dam number | 1°C | 2°C | 3°C | 4°C  | 5°C  |
|------------|-----|-----|-----|------|------|
| 1          | 2.2 | 3.5 | 9.2 | 17.0 | 31.9 |
| 2          | 1.8 | 2.0 | 3.7 | 7.3  | 12.2 |
| 3          | 1.6 | 2.6 | 4.3 | 6.9  | 12.1 |
| 4          | 1.7 | 2.9 | 5.2 | 7.7  | 14.1 |
| 5          | 1.6 | 2.4 | 3.4 | 5.3  | 7.4  |
| 6          | 2.1 | 3.8 | 6.3 | 11.9 | 19.2 |
| 7          | 1.9 | 2.6 | 4.8 | 8.2  | 15.0 |
| 8          | 1.0 | 1.0 | 1.7 | 2.7  | 7.5  |
| 9          | 1.2 | 2.7 | 3.1 | 5.7  | 7.7  |
| 10         | 1.3 | 1.6 | 2.0 | 2.5  | 3.2  |
| 11         | 1.6 | 2.5 | 3.8 | 6.0  | 9.1  |
| 12         | 1.5 | 2.2 | 3.3 | 4.6  | 6.1  |
| 13         | 1.7 | 1.8 | 2.9 | 5.1  | 7.6  |
| 14         | 1.4 | 1.7 | 2.3 | 3.2  | 4.5  |
| 15         | 1.7 | 1.6 | 2.5 | 4.2  | 7.5  |
| 16         | 1.8 | 3.5 | 6.6 | 13.8 | 24.7 |
| 17         | 1.5 | 2.3 | 2.9 | 5.2  | 8.6  |
| 18         | 1.2 | 1.4 | 2.0 | 2.4  | 4.6  |

**Discussion**

19. Line 367: the assumption made by the authors that climate change will lead to increasing the potential for dams to attenuate floods is not backed by the findings presented in this paper. I recommend to replace "will result" by "could result".

We have replaced "will result" with "likely result" given we follow up this statement with references to studies that have shown that climate change will increase demands for stored water resulting in lower initial reservoir levels.

20. Paragraph lines 384 to 399: authors justify the use of outputs from global climate models instead of regional ones, while in reality none of these outputs have been used here. The authors have simply applied a series of temperature increases to the hydrological drivers, without relying on any climate model. This could lead to misunderstanding the process followed. I recommend to replace this paragraph by a justification of the simplified methodology proposed.

We have added an explicit explanation in the third-to-last paragraph of the discussion of how our results, using covariates of global temperature, can then be related to various scenarios of climate change for any future time horizon as shown in Fig. 3. We have also clarified in the second-to-last paragraph of section 2.3 in "Materials and methods" that the rates of change used in our analysis are based on the results from previous studies that employed outputs from global and regional climate models to calculate the impact on hydrological drivers and their dependencies on global temperature change.

**Conclusion**

**21. Line 429: it is the first time that this result appears in the text. It should be mentioned before.**

This comment is in relation to the statement that "current levels of global warming relative to the period used to inform historic flood risk estimates in Australia mean that the risk of floods exceeding the dam crest flood is already more than twice as probable for four of the 18 dams investigated". Rather than having this result first appear in the conclusion, we have

introduced it in the second sentence of section 3. Results by noting the currently global temperature increase above the 1961-1990 baseline. We then presented the result in the paragraph above figure 5. We found an error in a data table used to specify the rate of change for storm temporal patterns and this has changed the number of dams with a doubled risk of exceeding the dam crest flood in the present day from four to two.

**22. Line 440: replace "practical approach" by "simplified approach".**

This comment is in relation to the final line: "Our study provides a practical approach for estimating extreme flood and dam overtopping risk under climate change that aligns with approaches widely used by practitioners making it feasible to be adopted globally." While we acknowledge that our implementation in representing rainfall and runoff for each dam was simplified, our method for assessing climate change on changes in the dam crest flood frequency, which is the primary focus of this paper, was not simplified. It was designed to be tractable in contrast to many existing top-down, scenario-driven methods of assessing climate change impacts on large floods. So, while it is, in comparison, simple, it is not a simplified approach. We have clarified this statement by adding: "Our study provides a practical and tractable approach"

**Technical corrections**

Unless a response is provided below, all the technical corrections have been made as recommended.

23. Line 42: rephrase "moisture delivered".

This has been revised to read "...timing of rainfall during a flood event"

- 24. Lines 56-57: consider revising the citation format.
- 25. Line 59: repetition "that that".
- 26. Line 89: reference for the Australian Rainfall and Runoff.
- 27. Table 1: instead of indicating dam owners, indicate dam type.

We have removed the column showing dam owners but have not included dam type as dam type is not relevant to the flood hydrology.

- 28. Line 169: replace "The rates of changed" by "The rates of change".
- 29. Figure 4: replace "notional overtopping" in legend by "DCF"
- 30. replace "Shift in overtopping risk" by "Shift in overtopping AEP"
- 31. y-axis does not represent the AEP, but the return period.

We have revised the x-axis tick marks to be the AEP as the 1/Y terminology is obstructing the clarity of the figure.

32. Figure 5: legend indicates that outliers are not shown, but it is the case in plots (b), (c) and (d)

Thank you for noticing this. The outliers should be shown for all plots including (a) and this has been corrected and the caption has been updated.

33. Line 325: change "appear" to "appears"

**Referee 2**

**General comments**

The topic is interesting and the manuscript is generally presented well. I found the consideration of temporal pattern innovative as this rainfall property is typically overlooked. Assumptions and methods are unclear though. Additionally, the results are not interesting/surprising and the discussion do not offer much insights. Here are my detailed comments:

1. Title: Add "Australia" to the title for clarity.

We have revised the title to "The impact of climate change on dam overtopping floods in Australia".

- 2. Graphical abstract: Write "Australia" on the map for clarity. Also, please clarify what variables you consider in your overtopping analyses (dam crest height, flood pool, maximum water level etc.) in the dam schematic (bottom right plot).
- "Australia" has been added to the map. We have added "reservoir level" to the y-axis of the figure (also recommended by Reviewer 1).
- 3. Abstract: Please clarify the future period and variables you consider in your overtopping analyses (dam crest height, flood pool, maximum water level etc.).

We did not limit our analysis to a specific future period as our analysis was dependent on degrees of global warming. We have clarified that overtopping floods refers to floods that exceed the dam crest flood.

4. Discussion of threats posed by aging dams (e.g., Ferdowsi et al. 2024; Shirzaei et al. 2025) may strengthen the introduction of your manuscript.

Thank you for these references – they have been included in our introduction. The article by Ferdowsi et al. (2024) provides a succinct argument for proactive dam safety assessments in response to climate change and increasing hydrologic extremes particularly in the Global South. We have referenced this article in the first paragraph of the introduction provide a better discussion of the global context. The article by Shirzaei et al. (2025) provides context for how climate change is but one component in dam risk assessments and disaster mitigation. We have revised our opening sentences to focus on dams, which is more appropriate for our paper.

5. Past research about dam overtopping, water level and inflow should be further acknowledged (e.g., Kwon & Moon 2006; Kuo et al. 2007; Hsu et al. 2011; Michailidi & Bacchi 2017; Wang & Zhang 2017; Cho et al. 2024, 2025).

Thank you for recommending these papers. I'm afraid at this stage we were unable to locate the article by Cho et al. 2025, but the remaining papers are all relevant to assessing dam overtopping floods and have been referred to in our introduction and discussion. We note that none of these papers consider climate change impacts on floods and therefore these references help underscore the novel contribution of our work.

6. The future period should be mentioned in the last paragraph of introduction section.

We have specified in the last paragraph of the introduction the range of global warming considered as we did not confine our analysis to a specific future period.

**7. Any reason for selecting 1961-1990 as the historic period and not a more recent period?**

Yes, we provide a justification for using this period as the historic period at the start of section 2.3. Namely: "The historic period approximates the mid-point for much of the information used to derive the design information provided in Australian Rainfall and Runoff (the national flood guidelines for Australia (Ball et al., 2019)), which establishes a baseline of historic flood risk with which to compare climate change impacts."

**8. L96: How did you classify the dams as large? Was it based on the ICOLD classification? Please clarify.**

We have clarified under "Section 2.1. Case study locations" that the 18 case study dams are classified as "large" based on the ICOLD classification of large dams as all the dams have a foundation to crest height exceeding 15 m. A reference to the ICOLD constitution that includes a definition of a large dam has been included.

9. The methodology is unclear. I suggest adding a schematic view of your overall methodology for estimating the dam overtopping. Also add a short write-up about how your methods connect to each other.

We have expanded on the schematic in Figure 2 to encompass the overall methodology by adding the step where the shift in overtopping flood frequency under climate change is considered. We have provided additional references to this figure throughout the description of the methodology to improve clarity on how the different method sections connect.

**10. Any reason for the selection of the 18 dams among other dams in Australia?**

The 18 dams are owned by authorities who expressed an interest in examining the change in their dam exposure to hydrological risk under climate change and provided support to this project with respect to financial support and sharing of data and models. The sharing of data and models is now stated in *Section 2.1. Case study locations*, while the financial support has been disclosed in the *Acknowledgements*.

**11. Are the 18 dams dependent on each other in terms of the inflows and water levels?**

These dams are all independent of each other with the exception of Somerset and Wivenhoe and a statement reflecting this has been included in Section 2.1.

12. Were the R2ORB models calibrated and validated? How did you account for changes in the land cover and surface roughness in the catchments? It would be helpful to discuss the changes in the historic period.

We have provided additional detail on how the R2ORB models were calibrated and validated to RORB models used in practice for design flood estimation in Appendix A. We have included our assumption that changes in land cover and catchment surface roughness were not considered as this was beyond the scope of the study. Furthermore, we are assessing relative changes resulting from climate change impacts rather than changes in catchment morphology over time. Please also see our response to reviewer 1, comment number 7.

13. Did you use the rainfall at the dam location or across the upstream catchment? The same question applies to antecedent soil moisture.

Design rainfall depths were applied at the approximate centroid of each catchment sub-area. The rainfall and antecedent soil moisture data used to inform the rates of change in initial and

continuing loss were based gridded data that were spatially averaged across the over 200 catchments used to inform the rates of change.

**14. Please show the study subcatchments in a supplement figure.**

Our preference is to not show the subcatchment configuration for the catchments to be consistent in our presentation across all the study sites. However, we note that our model configuration adequately captures the catchment behaviour as presented in our additional description of the validation of the models in Appendix A.

**15. How were future rainfall time series generated? Did you use any GCMs? Any bias correction and downscaling? Details are needed.**

We did not use projections of rainfall time series. We used a rate of change applied to a design rainfall depth described in section 2.3.1. The rate of change that we adopted is based on a published systematic review of observed historical trends, relationships between extreme rainfall and temperature, and findings modelled using both general circulation models and regional models. This additional detail has been added to section 2.3.1.

**16. PMP can be estimated via different methods. How sensitive your analyses with respect to the selected method?**

Our results suggested that the change in overtopping flood probability was most influenced by changes in rainfall depth. The historic estimate of PMP is therefore a baseline of our analysis and different estimates of PMP using different methods will consequently change our results, which is stated at the end of the second paragraph in *Section 2.3.1 Rainfall Depth*. The PMP method used here is based on generalised hydrometeorological methods as advocated by the WMO (2009), which is commonly used across the world and a reference to this has been included. It is worth noting that published studies of statistical estimates of PMP (e.g. Herschfield 1965, doi: 10.1002/j.1551-8833.1965.tb01486.x) yield similar results as hydrometeorological methods. These points are discussed in Visser et al (2022) and Wasko et al (2024), and we make reference to these in *Section 2.3.1. Rainfall Depth*.

**17. Rate of change factors were estimated based on climate zones. Would these remain stable under future climate?**

The rate of change factors for storm temporal patterns were based on Köppen-Geiger zones, while the rate of change factors for rainfall losses are based on Natural Resource Management region. The climate zones will likely change into the future (Beck et al. 2018). However, the rate change factors are based on trends in temporal patterns with current climate zones used by Visser et al. (2023) as a method for compositing and summarising results. We do not have information on how these scaling rates would change if the results were to be classified by future climate zone definitions. We have included this caveat when rate of change factors for temporal patterns are introduced in Section 2.3.2.

Beck, H. E., Zimmermann, N. E., McVicar, T. R., Vergopolan, N., Berg, A., and Wood, E. F.: Present and future Köppen-Geiger climate classification maps at 1-km resolution, Sci Data, 5, 180214, <a href="https://doi.org/10.1038/sdata.2018.214">https://doi.org/10.1038/sdata.2018.214</a>, 2018.

**18. Please discuss your methodology for combing AEPs of the three rainfall characteristics.**

Thank you for noting this omission. We have included a description of how the three flood characteristics under climate change are assessed individually and in combination in *Section 2.3. Assessing impacts of climate change*.

**19. 14a: Can you elaborate on the zone that shows a shift in overtopping risk?**

We assume this comment is in reference to Figure 4a showing the shift in the AEP of the overtopping flood between historic flood probabilities and those under 5°C of global warming. We have referenced the flood frequency curves corresponding to different degrees of global warming and related shifts in overtopping flood AEP within this zone in the paragraph under Figure 4.

**20. 14b: Can you show the temporal patterns of rainfall events as dimensionless time series showing the fraction of total rainfall?**

The figure attached shows the temporal patterns used for the different catchments. While this figure could be included in an appendix or Supplementary Material, we believe that providing these temporal patterns could distract the reader from the focus of the paper.

21. The term overtopping risk should be replaced with overtopping probability or hazard as you do not investigate the consequences of dam overtopping.

We have revised the terminology. Please refer to our response to Reviewer 1 point 2 for more detail.

**22. Please add a map that shows the overtopping probability under historic and future conditions.**

The shift in overtopping risks is a function of differences in catchment characteristics, configuration of a dam's outlet works, and hydroclimatic region. Proving a map of the differences only relates to one of these factors, and thus is potentially misleading. We have raised this point in the third paragraph of the Discussion.

**23. In general, I do not find any interesting results. Can you highlight the key findings of your paper?**

Our results show that on average dams are seven times more likely to be overtopped under a plausible climate scenario and we believe that these results are of considerable scientific and practical interest. To our knowledge this is the first time that projections have been systematically quantified for a wide range of hydroclimatic regions giving consideration to rainfall depths, antecedent conditions, and temporal patterns. This is both of scientific

interest, as the projections are based on our most current understanding of climate science as published in Wasko et al. (2024), and of direct relevance to industry, as they are based on models and procedures that are currently being used in engineering design practice. We have revised the abstract and conclusion to ensure that these aspects are highlighted.

**24. As acknowledged by the authors, the overtopping is a result of multiple factors acting together. As such, the reliability of this study is questionable.**

Our approach explicitly considers the joint probabilities of the factors impacting floods using the best available climate science. We agree that there are deep uncertainties associated with the trajectory of global warming and the dynamic factors that vary by location, and that there are additional uncertainties associated with hydroclimatic dependencies. That said, we need some basis to understand how climate change may impact on dam safety in the future. We consider the adopted methodology and derived results to be a useful contribution, despite the current irreducible uncertainties involved. These uncertainties have been documented (see response to comment 26), noting that such uncertainties undermine confidence in all such climate-impact studies.

**25. L245: Please revise the sentence.**

This original sentence was: "The catchments used in the study by Ho et al. (2023) were selected where a statistically significant relationship (at a significance level of a = 0.05) could be found between losses and antecedent soil moisture for 3-day rainfall events that were equalled or exceeded, on average, 5 times per year (a 5 EY event)."

This has been revised to: "The catchments included in the study by Ho et al. (2023) were those where a statistically significant relationship (at a significance level of a = 0.05) was found between losses and antecedent soil moisture."

**26. Sources of uncertainty and their impact on your results should be discussed.**

We believe that we have provided a discussion of the sources of uncertainty and likely impacts on our analysis in the discussion: We discuss plausible impacts of shifts in temporal patterns in the second paragraph, the assumption of reservoirs starting at a full supply level in the fourth paragraph, and the use of approximate central tendencies for the rates of change in the sixth paragraph.

**27. Section 5: Conclusion-> Conclusions**

This has been corrected as suggested.

---

## Author Response (AR2)

**Reviewer 1**

I appreciate the authors' efforts in addressing my comments. I see the manuscript is clearer now. Here are my few follow up comments:

1. On the selection of the 18 dams: You may want to add a statement about future research on studying a larger number of dams but also point out to the data availability and potential confidentiality issues.

We have now included this point in the third paragraph of the discussion as follows: "The feasibility of future studies that considers a larger sample size of dams will be dependent on the availability of data and confidentiality requirements related to dam operations.".

2. Here is the full reference of Cho et al. (2025): Cho E, Ahmadisharaf E, Villarini G, AghaKouchak A (2025) Historical changes in overtopping probability of dams in the United States. Nature Communications 16(1), 6693.

Thank you, this reference has now been included.

3. The rainfall and soil moisture should be based on the entire upstream watershed of each dam as opposed to a single point—the watershed centroid. This can be at least discussed as a limitation for future research.

We do not use a single point to represent rainfall and soil moisture for the catchments upstream of the dam. The catchments are divided into sub areas ranging from 4-19 sub areas as described in section "2.2 Event-based modelling". Rainfall is applied to the centroid of each sub area and rainfall excess is calculated at each sub area as described in Appendix A. The spatial distribution of rainfall can also be specified in the R2ORB model and we adopted the spatial weightings used by the dam owners.

4. Please also show the upstream watershed of each dam as a supplemental file (e.g., kmz) or map. It is unclear how large the upstream watershed of each dam was and why.

Table 1 shows the area of each catchment upstream of the dam and this provides clear information on the size of the catchment upstream of each dam.

We are unable to provide a map of the catchment upstream of each dam as this was not made available to us by the dam owners – we extracted the data we needed for our modelling purposes from their model files and ensured that our simulations agreed with their simulations. The most important point to note here is the area upstream of each dam, which we do provide in Table 1.

5. PMP was used as a baseline in your analyses. As I mentioned in my original comments, PMP can be estimated via different methods and changes your results. This was also acknowledged by the authors

in their response to my original comment. I think a sensitive analysis with respect to the selected method would add value to the paper results.

Our guiding principle with this study was to use methods and data sets that are consistent with accepted design practice in Australia – that is, these results are based on "real world" practice. While we accept that the results may be sensitive to various methodological aspects – the PMP method just being one – such an exercise would greatly expand upon the scope of the work. Hopefully this work will encourage researchers in other countries to explore the implications using different data sets and methods.

More specifically, we note that there is only one accepted method of estimating PMPs in Australia for each region, which we have used in this study (as stated in Section 2.3.1). But regardless, one reason we do not think the choice of PMP method is of great concern is that our focus is on the relative shift in overtopping probabilities over a range that lies between statistical estimates derived from the observed record and those inferred from PMP methods. Different PMP methods are associated with different exceedance probabilities, and thus while a different PMP estimate might shift the absolute estimate overtopping probability, there is no compelling reason to expect that it would influence the relative shift from one climate scenario to another.

6. Please present the temporal patterns of rainfall events as dimensionless time series showing the fraction of total rainfall as a supplement figure.

We have now included the figure that previously appeared in our response to the reviewer in the supplementary material.

7. I do not understand why adding a map of overtopping probability under historic and future conditions can be misleading. You present changes in the probability in the text as numbers. How can a map, which aggregates the same information in one figure, can be misleading?

We were unable to find any useful relationship between changes in overtopping flood risk under climate change and geographical location (as stated in the third paragraph of the discussion). We found that mapping the results has led audiences to propose the existence of a geographical influence, likely due to the spatial distribution of the case studies that are largely clustered in the south-east of Australia. However, this geographical influence is not statistically supported, which is our reason for demurring on the suggestion to present the results geographically.

**Reviewer 2**

I have carefully reviewed the revised version of the manuscript and am pleased to see that it has improved significantly.

I suggest some minor technical corrections:

- Lines 44-45: cite more recent works.

Thank you for the suggestion. We have added references to the works of Cho et al., 2025 and Rajabzadeh et al., 2023.

- Lines 140 and 421: "social, economic, and political factors"

**The text has been revised as suggested.**

- Line 199: replace "calculated" by "calculate"

The text has been revised as suggested.

---

## Author Response (AR3)

I thank the authors for their revisions and am glad to accept the paper for publication, subject to the amendment of two minor/technical points:

- In response to a comment by Referee 1 (#6 in your response), you indicated that supplementary material was added. However, I was unable to locate it in the uploaded files—please complete your submission if it was missing.

Apologies, our response with respect to the inclusion of an additional figure suggested by Referee 1 showing the temporal patterns should have read:

"We have now included the figure that previously appeared in our response to the reviewer in Figure A 1 in Appendix A." rather than "supplementary material".

- In addition, please ensure that the new supplementary figure is referenced in the main text.

The new figure, Figure A 1 in Appendix A, has now been referenced in L257-258.

- Lines 238–240, please fix the reference

The reference to Roderick et al. 2020 has now been fixed.

Thank you